# The Effect of Barley and Lysine Supplementation on the *longissimus lumborum* Meat Quality of Pasture-Raised Fallow Deer (*Dama dama*)

**DOI:** 10.3390/foods9091255

**Published:** 2020-09-08

**Authors:** Daniel Bureš, Luděk Bartoň, Eva Kudrnáčová, Radim Kotrba, Louwrens C. Hoffman

**Affiliations:** 1Institute of Animal Science, 104 00 Prague 10-Uhříněves, Czech Republic; barton.ludek@vuzv.cz (L.B.); kudrnacovae@af.czu.cz (E.K.); kotrba.radim@vuzv.cz (R.K.); 2Department of Food Science, Faculty of Agrobiology, Food and Natural Resources, Czech University of Life Sciences, 165 21 Prague, Czech Republic; 3Department of Animal Science and Food Processing, Faculty of Tropical AgriSciences, Czech University of Life Sciences, 165 21 Prague, Czech Republic; 4Centre for Nutrition and Food Sciences, Queensland Alliance Mechanisation Building A. 8115, Gatton 4343, Australia; louwrens.hoffman@uq.edu.au; 5Department of Animal Sciences, University of Stellenbosch, Private Bag XI, Matieland 7602, South Africa

**Keywords:** amino acids, bucks, diets, nutrition, sensory analysis, venison

## Abstract

The chemical characteristics (proximate composition, amino acids, and fatty acids) and sensory quality of the *longissimus lumborum* (LL) muscle of 45 farmed male fallow deer were investigated. The animals were divided into three separate groups (*n* = 15 per treatment): pasture-fed (P), pasture-fed and supplemented with barley (B), and pasture-fed and supplemented with barley and lysine (BL). Differences were observed in LL moisture and the intramuscular fat contents, the latter being almost two-fold greater in the meat of B and BL groups compared to P. The concentrations of histidine, leucine, alanine, glutamic acid and glycine in the raw meat were higher in the BL group compared to the P group. Higher contents of *n*–3 polyunsaturated fatty acids (PUFAs), and consequently lower *n*–3 ratios, were found in the P group, compared to the BL group. The grilled meat samples from the P group scored higher than the other groups for grassy flavour, and lower for liver flavour.

## 1. Introduction

Deer species have been utilized by humans world-wide for hunting, and for the production of meat and other animal products, for centuries. Deer meat, or venison, has a long history in both western and eastern cultures, but it is also currently rather high in terms of consumer demand [1]. For example, in the Czech Republic, the evaluated data on meat consumption report a game meat per capita consumption of 1.0 kg/person/year, out of a total meat consumption of 80.3 kg/person/year. Although this figure may seem low in comparison with the total meat consumed, the consumption of game meat has more than tripled over the last fifteen years in the Czech Republic. It is clear that this type of meat is becoming more readily available in the market for ordinary consumers [2]. Venison may represent a favourable food commodity for consumers, because it is lean and tasty, with a high proportion of polar lipids and proteins, and therefore perceived positive effects on human health [3]. Additionally, due to the ethical nature of production and restricted use of pharmaceuticals, venison has a positive consumer image as a natural, “organic”, and safe product [4]. Farming deer also has the potential to provide a regular supply of uniform quality meat to the commercial market, in comparison to the meat generated through hunted deer [1,3].

Deer-keeping under farm conditions is predominantly characterized by pasture fattening. However, grazing or browsing alone may be insufficient in meeting their nutritional requirements, especially during the period of the year when grazing is poor, or when the stocking density exceeds the carrying capacity, and thus supplementary feed is often supplied to farmed deer [5,6,7]. Supplementary feeding can also improve animal welfare, growth intensity, carcass composition, and influence meat quality. In general, deer fed or supplemented with concentrates have higher levels of intramuscular fat (IMF) [6,8], lower concentrations of polyunsaturated fatty acids (PUFAs) to saturated fatty acids (SFAs), as well as PUFA n–3:PUFA n–6 ratios [5,7], than animals without supplementary feeding. Furthermore, meat from fallow deer (FD) does that were fed concentrates exhibited significantly stronger flavour attributes compared to animals which only grazed pastures [8].

The efficiency of metabolizable proteins used by ruminants is highly dependent on the profile of absorbable amino acids (AAs) in the feed, because the deficiency of a single AA can limit the use of other AAs that are in adequate supply [9]. Lysine is often regarded as a first-limiting AA in growing ruminant diets [10] for protein biosynthesis [11]. Feed commonly used for ruminants tends to meet the requirements for rumen-degradable protein, but are often deficient in rumen-undegradable protein. Therefore, dietary supplementation is often necessary, through the inclusion of AAs protected from ruminal degradation [12], known as rumen-protected amino acids (RPAAs). It is essential to provide these AAs in this protected form so that they can withstand microbial degradation in the rumen, and ensure availability in the abomasum, where they are more efficiently utilized. A recent study showed that the inclusion of lysine and methionine in deer feed rations increases growth performance, while decreasing environmental waste and economic costs [13]. The effects of increasing dietary lysine levels on lipogenesis inhibition and the repartitioning of nutrients towards lean tissue deposition, instead of towards fat deposition, have been reported previously in monogastric farm animals, such as chickens [14], ducks [15] and pigs [16]. However, in previous work on FD, it was found that the supplementation of FD bucks with lysine also significantly decreased internal fat and carcass fat deposition [17]. Despite the fact that FD are one of the most popular deer species farmed world-wide, few studies have evaluated the effects of feeding concentrate and amino acid supplementations in this species. Therefore, a considerable amount of information is still missing in this research area, particularly pertaining to the effect of dietary changes on FD meat quality. Thus, the purpose of this study was to describe the effect of feeding supplementary diets to fallow deer on the chemical composition (proximate, amino acids, and fatty acids) of the *longissimus lumborum* muscle and its organoleptic quality, by means of a descriptive sensory analysis.

## 2. Materials and Methods

### 2.1. Animals, Experimental Design

The study was approved by the Ethic Committee of the Institute of Animal Science (IAS; No. 60444/2011-MZE-17214). The feeding experiment was conducted at a commercial deer farm in Mnich (49.1672144N, 14.9007917E) within the Czech Republic. A total of 45 fallow deer males were included in the experiment, at an average initial age of 11 months and live weight of 28.2 ± 1.8 kg. The animals were randomly assigned to three diet treatment groups, with 15 animals per group, and placed in three 2 ha adjoining paddocks. The pasture predominantly consisted of *Lolium perenne* and *Cynosurus cristatus*, and naturally occurring grass species of *Agrostis* sp., *Festuca* sp. and *Poa* sp. *Trifolium repens*, and the weed species *Urtica dioica* and *Cirsium arvense*, were also identified in the paddocks. The pastures were regularly cut and, after drying, also partially consumed by deer. The bucks were weighed three times during the experiment, and the groups were rotated amongst the paddocks at six-week intervals, to minimise the effect of the “paddock”. Group P received only pasture, while group B received pasture supplemented with barley, and group BL received pasture supplemented with barley, which was mixed with RPAA lysine (LysiPEARL^TM^, Kemin Industries, Inc., Des Moines, IA, USA) at a level of 5 g/day/animal. This preparation provided 2.5 g lysine per animal daily, in an encapsulated form. The experimental period was divided into two phases: in the first phase (90 days), the B and BL groups were supplemented with barley at 0.2 kg/day per animal, whereas in the second phase (~79 days), barley supplementation was increased to 0.4 kg/day per animal. Further details regarding the experimental design, chemical and nutritional compositions of the feeds used, as well as the growth performance, carcass composition and physical characteristics of the *longissimus lumborum* and *semitendinosus* muscles of these animals have been reported [17]. A brief summary of the key results is reported in Table 1, to give a general overview of these parameters.

### 2.2. Slaughter and Muscle Sampling

The experiment was terminated by slaughtering the animals at the farm (no transport) over three days, at an average age of 17 months and after 169 days of feeding, following procedures typically applied to deer in the Czech Republic. On each of the three slaughter days, a captive bolt was used to stun 15 animals (5 randomly selected from each group) whilst contained in a handling box, after which the FD were immediately bled and eviscerated, and then transferred in a refrigerated truck to the experimental slaughterhouse of IAS. Here, the carcasses were uniformly dressed and stored at +4 °C until further processing. Ninety-six hours (four days) after slaughter, the whole *longissimus lumborum* (LL) muscles were removed from both carcass sides. The muscles were cleaned of subcutaneous fat and epimysium, and were transported in a cooling box to the laboratory for chemical and sensory analyses.

### 2.3. Proximate Chemical Composition, Amino Acids and Fatty Acid Analyses

The LL samples (approximately 300 g, from the right carcass sides) intended for chemical analyses were homogenized and frozen at −20 °C until analysis. The dry matter content was determined by oven-drying at +105 °C to a constant weight. Dried samples were pulverized using a Grindomix GM 200 knife mill (Retch, Haan, Germany) and analysed for crude protein content using a Kjeltec 2400 Analyzer (FOSS Tecator AB, Höganäs, Sweden). The intramuscular fat content was determined after extraction with petroleum ether using a Soxtec Avanti 2055 System (FOSS Tecator AB, Höganäs, Sweden). The collagen content was determined according to a methodology adapted from earlier work in the respective laboratory [18]. Soluble (heat-labile) collagen was determined by measuring the hydroxyproline content from duplicated 1 g samples. The total collagen was expressed as g/100 g of muscle tissue. The collagen solubility was expressed as the percentage of heat-soluble collagen within the total collagen fraction. The amino acid content of the homogenized and dried LL samples was determined with an AAA400 Automatic Amino Acid Analyzer (Ingos, Prague, Czech Republic), with Na-citrate buffers and ninhydrin detection. The samples were hydrolysed with 6 M HCl at +110 °C for 23 h. For analysis of cysteine and methionine, the samples were oxidized with a mixture of 85% formic acid and 30% hydrogen peroxide (9:1) prior to hydrolysis. Tryptophan was not determined, as it is completely destroyed during acid hydrolysis. The fatty acid (FA) composition of the LL muscles was determined after the extraction of total lipids, according to Folch and colleagues [19]. After the alkaline transmethylation of the FAs, gas chromatography was performed with a HP 6890 gas chromatograph (Agilent Technologies, Inc., Santa Clara, CA, USA) using a 60 m DB-23 capillary column (150 to 230 °C). The FAs were identified on the basis of retention times corresponding to standards (PUFA 1, PUFA 2, PUFA 3, CRM-Supelco 37 Component FAME Mix; Supelco, Bellefonte, PA, USA). The proportions of FAs were expressed as percentages of the total area of the injected methyl esters.

### 2.4. Descriptive Sensory Analysis (DSA)

All samples intended for DSA (the whole LL from the left side, and the caudal half of the LL from the right side) were vacuum packaged and refrigerated at 4 °C for a further 10 days (a total of 14 days after slaughter). The samples were then frozen and stored at −20 °C for 60 days, until the start of the DSA. Frozen LL samples were thawed for 24 h at 4 °C prior to sensory evaluation, then cut into 2 cm steaks, perpendicular to the length of the LL muscle. The steaks were prepared by grilling them on a preheated (200 °C) double glass/ceramic plate grill (VCR 6L TL, Fiamma, Aveiro, Portugal), until an internal temperature of 70 °C was reached (digital temperature probe AD14TH, Ama-Digit, Kreuzwertheim, Germany). Immediately after cooking, samples were cut into 2 cm^3^ blocks (excluding the outer meat surface), placed into glass containers, and sealed. Each container/sample was marked with a randomized three-digit code, and placed into an oven at 50 °C for 1 h until evaluation. Subsequently, the DSA of the samples was performed in individual booths, under controlled environmental conditions and red light, using ten trained and experienced panellists, during a total of three sessions. Five sets of samples were presented to each panellist in each of these three sessions. All sets consisted of three samples, one from each of the three experimental groups slaughtered on the same day. Thus, the sensory analysis included all samples of the 45 animals slaughtered, evaluated in a total of 15 sets. To avoid possible effects of the order of presentation and first-order carryover effects, the sample sets were presented to panellists in different randomized orders. Panellists were provided with water and bread to cleanse their palates between samples. General training was performed during two previous sessions, using FD meat, for the understanding and development of the descriptors under consideration. A linear unstructured continuous 100 mm scale, oriented by descriptions at both ends, was used for each of the nine descriptors defined in Table 2. This scale was then transformed into a numerical scale (0–100) for statistical analysis.

### 2.5. Technological Parameters

Thawing losses were calculated as the difference between the sample weights before freezing and after thawing. Cooking losses were calculated as the differences between the sample weights before and after grilling.

### 2.6. Statistical Analysis

The statistical analysis was performed in the statistical package SAS (Version 9.4, SAS Institute Inc., Cary, NC, USA). Variables were tested for normality using the Shapiro–Wilk test, and for homogeneity of variance with Levene’s test. Data were subsequently analysed with a mixed linear model, and the parameters were estimated by the REML method in the MIXED procedure. The statistical model for chemical composition of LL samples involved the fixed effect of “diet” and the random effect of the “slaughter day”. The model used for evaluating the sensory characteristics included the fixed effect of “diet” and the random effects of the “slaughter day” and “assessor”. For post-hoc analyses, Tukey’s range tests were used. Differences were considered significant at the level of *p* < 0.05.

## 3. Results

### 3.1. Proximate Chemical Composition, Amino Acid Content, and Fatty Acid Proportions of the LL Muscle

The results of the proximate chemical composition of the LL are presented in Table 3. The moisture content was 0.65 and 1.06% higher in the meat from P animals than in the meat from B and BL animals, respectively (*p* < 0.001). No differences were observed for the LL crude protein content between treatments, whereas both supplemented groups (B and BL) had crude protein contents of 97.5 and 85.0%, respectively, and higher contents of IMF in their LL than the P group (*p* < 0.001). The total collagen content tended to be 12.5 and 9.1% higher in the meat from the B animals, compared to the P and BL groups, respectively (*p* = 0.061). Collagen solubility tended (*p* = 0.077) to be higher in the meat from the P group, compared to the other groups.

The AA content of the *longissimus lumborum* muscle from the various diet treatments are given in Table 4. Only a few of the AAs differed significantly between the dietary treatment groups. The leucine, alanine, glutamic acid, and glycine contents were higher (*p* < 0.05) in the BL meat compared to the P meat, by 3.4, 4.2, 3.9, and 3.4%, respectively. However, the concentrations of histidine, glutamic acid and glycine were higher (*p* < 0.05) in the BL meat than in the B meat, by 3.3, 3.2, and 3.4%, respectively. In addition, the concentrations of isoleucine, phenylalanine, cysteine and serine tended to be higher (*p* < 0.10) in the BL meat compared to the other treatment groups. No significant differences were observed between the P and B groups for any of the AAs.

The proportions (g/100 g of FAs determined) of fatty acids (analysed in the muscle) are presented in Table 5. Only major FAs exceeding 0.1 g/100 g of the total FAs, and representing more than 99% of total FAs, are reported. Only a few significant differences between the treatment groups were detected for individual FAs. The proportion of eicosapentaenoic acid (EPA; C20:5 *n*–3) present was 17.6% higher in the muscle of P animals than in the muscles from the BL group, whereas the proportion of docosahexaenoic acid (DHA; C22:6 *n*−3) was 13.9% higher in the muscle of P animals compared to the muscles from the B group. As a result of the generally higher individual *n*−3 PUFA in the pasture-fed (P) group, the proportion of total *n*−3 PUFA was 16.1% higher in the muscle from P deer than in the BL (*p* < 0.01) deer, and the ratio of *n*−6 PUFA/ *n*−3 PUFA was 9.5 and 10.9% lower in the P muscles, compared to the muscles from B and BL groups, respectively (*p* < 0.05).

### 3.2. Descriptive Sensory Analysis

The results of the descriptive sensory analysis of the grilled LL muscles are given in Table 6. No significant differences were observed for the aroma (aroma intensity and game aroma intensity) and texture (tenderness and juiciness) characteristics. Of the variety of flavour characteristics analysed, the taste panel perceived differences between the dietary treatment groups for grassy and liver flavour attributes. The P meat samples received higher scores for grassy flavour (*p* < 0.05), and lower scores for liver flavour (*p* < 0.01), than the meat samples from the concentrate-supplemented groups (B and BL). The addition of lysine to the barley supplementation diet had no significant effect on the sensory characteristics of grilled LL when compared with the meat from the animals that were only supplemented with barley. The weight losses associated with the thawing and cooking processes of the meat are shown in Table 6. The lowest thawing loss was reported for the P group (*p* < 0.001), with the values of the B and BL groups being higher than that of the P group by 2.8 and 2.5 percentage points, respectively. On the other hand, the cooking loss differences were small and insignificant between dietary treatments.

## 4. Discussion

This experiment aimed to evaluate the effect of diet on FD bucks’ LL muscle chemical composition, including amino acid and fatty acid contents, and organoleptic properties. Animals were reared under identical conditions, and entered the experiment at a similar age and live weight. To the best of our knowledge, this study is the first contribution towards describing the effects of rumen-protected lysine on the chemical composition and sensory parameters of FD meat. Furthermore, no available studies have yet been published reporting the AA content of FD meat, or the effects of rumen-protected lysine supplementation thereon.

### 4.1. Proximate Chemical Composition, Amino Acid and Fatty Acid Content

Both supplemented groups had almost two-fold higher contents of IMF in their LL muscle, and consequently had lower moisture contents, compared to the pasture-fed FD, which is consistent with the results of another feeding experiment using concentrate-fed FD [6]. In a previous study, it was observed that the proportions of the carcass and internal fat depots were lower in lysine-supplemented FD compared to those fed barley [17].

The IMF levels in the meat from B and BL groups were similar to those reported in farmed FD bucks slaughtered at the same age and supplemented with a mixture of oats, barley and maize [20]. In contrast, lower IMF values (2.4 g/kg muscle) for farmed FD supplemented with a cereal mixture, and slaughtered at 17–18 months of age have been reported [21]. However, other authors reported greater IMF values in FD animals who were older at slaughter, with this likely being attributed to various factors such as age, diet and season [8,22,23]. Additionally, the lipid extraction methodologies used differed between the studies discussed, which may have also attributed to the differences observed between findings.

Typically, the crude protein content of uncooked *longissimus* muscle from other ruminant game species ranged from 20 to 24 g/100 g [24]. In the present study, the muscle crude protein contents ranged from 23.39 g/100 g in the P group, to 23.55 g/100 g in BL animals, and were slightly higher than those reported by other studies [6,20,22]. Consistent with these findings, the muscle protein content for deer supplemented with concentrates was reported to be only numerically, and nonsignificantly, higher than the protein content of muscles from animals with only access to pasture [6].

Collagen is a major component of the intramuscular connective tissue, and its content and composition are responsible for the background toughness of cooked meat [25]. In the present study, and in agreement with previous findings [6], supplementary feeding had no influence on the total collagen content nor collagen solubility in FD meat.

Venison is an important source of dietary AAs for humans, used to sustain adequate protein nutrition and health [26]. Lysine and leucine (essential AAs), as well as glutamic and aspartic acids (nonessential AAs), were considered as major AAs in the LL muscle. From available sources focusing on the AA composition in cervid muscles, data are provided for roe deer (*Capreolus capreolus*) [27] and red deer (*Cervus elaphus*) [28,29]. In agreement with the present study’s results, lysine and leucine were the most abundant essential AAs in these studies on other deer species. It has also been noted that the AA profile does not vary greatly between species [30]. Indeed, similar AA profiles were reported in the LL from eland (*Taurotragus oryx*) and cattle (*Bos taurus*) [31], as well as from impala (*Aepyceros melampus*) [32] and kudu (*Tragelaphus strepsiceros*) [33]. The slight increase in several muscle AAs (Table 4) may have been due to the more efficient use of the available feed nutrients in muscle protein production. However, the underlying cause of the effect of feeding RPAAs on the AA composition in meat is not clear and requires further investigation. For example, in beef cattle, it was found that feeding rumen-protected lysine increased the carcass mass and leanness [34]. Increased dietary (protected in the case of ruminants) lysine increases muscle anabolism, particularly during the early growth stages of ruminants [34,35]. In lambs, a 3.7% increase in nitrogen retention was noted when the animals were fed rumen-protected lysine [36]. Although others have found that lysine supplementation had no effect on the carcass parameters evaluated [37,38], it is thought that in these latter studies adequate levels of lysine were already provided in the cereal-based diets or that the animals had already entered into a slower anabolic growth phase. The stage at which protein anabolism starts decreasing in deer is unknown and warrants further research. In pigs, it was also noted that lysine is not only an important building block in muscle accretion, but it also acts as a signal that activates stem cells to manipulate muscle growth via the mammalian target of the rapamycin complex 1 (mTORC1) pathway [39]. It should be remembered that even though additional dietary lysine, when this amino acid is limiting and the animal is in an anabolic state, will increase muscle accretion, the probability remains that the proportion of lysine and the other amino acids to each other will remain the same when expressed as mg amino acid/protein. This is because muscular protein tends to have the same amino acid composition, as noted earlier [30,31,32,33].

In the current study, the most abundant FAs in FD meat were oleic (C18:1 *n*–9), palmitic (C16:0), stearic (C18:0), and linoleic (C18:2 *n*–6) acids. These results were consistent with previous findings [20], and within other studies investigating the FA profile of meat from feral and farmed FD [21,40]. As reviewed, dietary lipids are extensively biohydrogenated in the rumen by ruminal micro-organisms, but diet is still the major environmental factor influencing the FA composition of meat from ruminants [41]. Forages, a rich source of linolenic acid (C18:3 *n*−3), can be used to increase the *n*−3 PUFA content of meat from ruminant species, relative to cereal-based diets. Indeed, in the present study, the higher concentration of dietary linolenic acid contained in the pasture-only diet may have contributed to the higher *n*−3 PUFAs, and the lower *n*−6 PUFA/*n*−3 PUFA ratio, found in the meat of the P group. In agreement with this, higher *n*−3 PUFA and lower *n*−6 PUFA/*n*−3 PUFA levels were reported in the *longissimus thoracis et lumborum* and *semitendinosus* muscles of male fallow deer receiving a pasture diet, compared to those supplemented with concentrates [6]. Similarly, higher *n*−3 PUFA and lower *n*−6 PUFA/*n*−3 PUFA levels were observed in pasture-fed, compared to commercial pellet-fed, red deer [5]. Regarding human nutrition, a more balanced intake of *n*–6 PUFAs and *n*−3 PUFAs in western diets is currently recommended, and the dietary *n*−6 PUFA/*n*−3 PUFA content should ideally be below five [42]. The pooled mean value of the FD meat *n*−6 PUFA/*n*−3 PUFA ratio (2.56) found in the present study was well below this threshold. Higher proportions of EPA and DHA observed in the meat from the P group may have been due to the enzymatic elongation and desaturation of linolenic acid to its longer chain derivatives [43]. A sufficient intake of very long chain *n*−3 PUFAs in the human diet is important, to ensure the optimal conditions for growth, development, and the maintenance of health [44], making FD an ideal food source.

### 4.2. Descriptive Sensory Analysis

Eating quality is a combination of tenderness, flavour and juiciness, and it is affected by numerous factors [45]. In grass-feeding systems, it is not possible to standardize all of these factors. Therefore, the interpretation of the results from such studies and comparisons with other experiments are challenging [46]. In the present experiment, the LL muscles were aged for fourteen days from the day of slaughter, frozen, and then defrosted and grilled before sensory evaluation. Interestingly, the lower thawing losses observed in the meat from the pasture-based group P were not reflected in the texture characteristics of the meat. This phenomenon may be attributed to the higher pH typically found in the meat of these “wild” animals [19]. The panellists were unable to detect any significant differences amongst the groups for aroma and texture characteristics, but they observed distinctions in two flavour characteristics. The results indicated that venison from concentrate-fed FD scored higher in liver flavour, whereas a greater intensity of grassy flavour was detected in the meat from grass-only fed animals. Panellists were not able to detect any differences between meat from pasture-fed and concentrate-fed FD regarding texture characteristics, in spite of their differing IMF contents, which is often positively associated with these sensory properties [47]. However, textural differences were not expected in the present study, for a number of reasons: there were no differences in the meat collagen content (Table 3) between diets, the animals were of the same age and sex, and the LL muscle samples were subjected to the same freezing/defrosting treatment. Similar to the present study, the sensory attributes of meat from reindeer (*Rangifer tarandus tarandus*) were reported to have a significantly greater liver flavour in the meat from animals fed a commercial feed mixture, compared to those which grazed on natural pasture [48]. A significantly greater grassy flavour was also detected in venison from grazing-only red deer, in comparison with the animals fed a commercial pelleted feed [5]. It is likely that an increased liver flavour or fatty flavour intensities contribute to a stronger and more desirable flavoured meat, as a result of concentrate-feeding [49]. However, as reviewed, each consumer’s preference between different flavours is very individual and depends on various factors such as culture or previous experiences [46]. Meat flavour can also be affected by FA composition [50]. It is well-recognized that cattle reared on grass produce a grassy flavour in their meat, due to naturally high levels of linolenic acid and EPA. These differences in FAs give rise to different profiles of aroma compounds during cooking [51]. A grassy, or pastoral, aroma and flavour is also influenced by the product of α-linolenic acid oxidation, and its derivatives (hexanal) [46]. A grassy or green aroma has been linked to 1-penten-3-ol, as well as selected aldehydes, such as hexanal (green-grassy, green-apple), octanal (green-fresh) and nonanal (green) [52,53].

## 5. Conclusions

The results of the present study demonstrate the possibility of modifying various meat quality characteristics of the *longissimus* muscle from FD through nutrition. While the addition of the dietary RPAA lysine changed the amino acid composition of the muscle slightly, the supplementary feeding of barley increased the IMF content and decreased *n*−3 PUFA concentrations. The grilled meat from grass-fed animals were distinguished by a grassy flavour, whereas the meat from supplemented FD had higher liver flavour scores. The results confirm that the meat of farmed FD can be considered as a nutrient-dense foodstuff, high in protein, low in intramuscular fat, and with suitable amino acid and fatty acid compositions. It is also characterized by favourable organoleptic properties; in particular, acceptable textural characteristics and outstanding flavour properties, even though the fat content is low. Further research is required to determine whether other AAs, including lysine supplementation, could improve the quality of venison, as consumers consider meat from deer species to be healthy alternatives to traditional farmed species.

## Figures and Tables

**Table 1 foods-09-01255-t001:** Means ± standard deviations of the growth performance and carcass traits of fallow deer bucks receiving different diets (pasture only, barley + pasture, barley + lysine + pasture). For further details regarding supplementation rates and diet composition, see [17].

	Diet		
Parameter	Pasture (P)	Barley (B)	Lysine (BL)
Number of observation/animals	15	15	15
Initial weight (kg)	28.4 ± 1.9	28.5 ± 1.4	27.8 ± 2.0
Slaughter weight (kg)	45.3 ± 1.8	50.5 ± 2.8	49.8 ± 3.4
Daily gain—over entire experiment (g/day)	100.7 ± 13.1	131.0 ± 15.1	130.1 ± 13.0
Carcass weight (kg)	23.0 ± 3.0	28.4 ± 3.0	27.8 ± 2.8

**Table 2 foods-09-01255-t002:** Definition and scale of attributes used in sensory analysis of farmed fallow deer meat.

Attribute	Evaluation	Definition	Scale
Aroma intensity	Before eating	The overall intensity of aroma	0 = cannot be identified 100 = extremely strong
Game aroma intensity	Before eating	Aroma associated with meat from wild animal species	0 = cannot be identified 100 = extremely strong
Tenderness	After first two or three chews	The force required to bite through the sample with molars	0 = very tough 100 = very tender
Juiciness	After first three to five chews	The amount of moisture released by the sample	0 = very low 100 = very high
Flavour intensity	After first five to ten chews	The presence of a flavour typical for cooked meat	0 = cannot be identified 100 = extremely strong
Game flavour intensity	After first ten to fifteen chews	Flavour associated with meat from wild animal species	0 = cannot be identified 100 = extremely strong
Grassy flavour	After first ten to fifteen chews	Fresh, clean, green, grass-fed flavour	0 = cannot be identified 100 = extremely strong
Astringency flavour	After first ten to fifteen chews	Dry, mouth-puckering, sharp, metallic, bitter	0 = cannot be identified 100 = extremely strong
Liver flavour	After first ten to fifteen chews	Flavour associated with pan-fried liver	0 = cannot be identified 100 = extremely strong

**Table 3 foods-09-01255-t003:** Least Square Means (LSM) and Standard Error (SEM) of the chemical composition (g/100 g muscle) of fallow deer *longissimus lumborum* muscle, after feeding with three different diets.

	Diet				
	Pasture (P) LSM	Barley (B) LSM	Lysine (BL) LSM	SEM	*p*-Value
Moisture	74.7 ^a^	74.2 ^b^	73.9 ^b^	0.17	<0.001
Crude protein	23.4	23.5	23.6	0.21	0.384
Intramuscular fat	0.40 ^b^	0.79 ^a^	0.74 ^a^	0.046	<0.001
Total collagen	0.32	0.36	0.33	0.016	0.061
Soluble collagen (%)	44.7	42.3	42.4	1.81	0.077

^a,b,c^ Means with different superscripts within the same row differ significantly (*p* < 0.05).

**Table 4 foods-09-01255-t004:** Amino acid composition (g/100 g muscle) of the *longissimus lumborum* muscle from fallow deer fed different diets.

	Diet				
	Pasture (P) LSM	Barley (B) LSM	Lysine (BL) LSM	SEM	*p*-Value
*Essential*					
Arginine	1.33	1.34	1.35	0.035	0.316
Histidine	0.92 ^ab^	0.92 ^b^	0.95 ^a^	0.022	0.039
Isoleucine	1.02	1.02	1.05	0.015	0.072
Leucine	1.74 ^b^	1.76 ^ab^	1.80 ^a^	0.033	0.023
Lysine	1.78	1.75	1.79	0.050	0.358
Methionine	0.59	0.59	0.61	0.005	0.044
Phenylalanine	0.89	0.88	0.91	0.023	0.059
Threonine	0.96	0.96	0.98	0.023	0.188
Valine	1.08	1.10	1.11	0.027	0.295
*Nonessential*					
Alanine	1.19 ^b^	1.22 ^ab^	1.24 ^a^	0.023	0.015
Aspartic acid	1.98	1.99	2.05	0.042	0.035
Cysteine	0.19	0.19	0.20	0.003	0.055
Glutamic acid	3.08 ^b^	3.10 ^b^	3.20 ^a^	0.090	0.010
Glycine	0.87 ^b^	0.87 ^b^	0.90 ^a^	0.027	0.011
Proline	0.77	0.78	0.79	0.019	0.328
Serine	0.78	0.79	0.80	0.015	0.053
Tyrosine	0.96	0.96	0.95	0.029	0.763

^a,b,c^ Means with different superscripts within the same row differ significantly (*p* < 0.05).

**Table 5 foods-09-01255-t005:** Fatty acid proportions, sums and ratios (g/100 g of total fatty acid determined) of fallow deer receiving different diets.

	Diet				
	Pasture (P) LSM	Barley (B) LSM	Lysine (BL) LSM	SEM	*p*-Value
C14:0	1.34	1.37	1.37	0.172	0.958
C14:1 *n*−5	0.10	0.10	0.11	0.012	0.749
C15:0	0.36	0.43	0.40	0.047	0.210
C16:0	22.46	22.59	23.25	0.382	0.298
C16:1 *n*−7	1.70	1.75	1.96	0.158	0.487
C17:0	0.53	0.57	0.57	0.059	0.671
C18:0	13.55	13.26	13.60	0.679	0.677
C18:1 *trans*−9	0.26	0.24	0.26	0.019	0.492
C18:1 *trans*−11	0.48	0.48	0.43	0.035	0.431
C18:1 *n*−9	30.87	30.96	31.33	1.373	0.832
C18:1 *n*−7	2.13	2.06	2.11	0.068	0.771
C18:2 *trans*−6	0.07	0.07	0.07	0.009	0.899
C18:2 *n*−6	13.08	13.60	12.80	0.386	0.341
C18:3 *n*−6	0.19	0.19	0.17	0.015	0.233
C18:3 *n*−3	3.07	2.86	2.59	0.272	0.116
*cis-*9 *trans*-11 CLA ^a^	0.13	0.11	0.12	0.010	0.387
C20:0	0.13	0.13	0.12	0.018	0.580
C20:1 *n*−9	0.24	0.23	0.22	0.021	0.678
C20:2 *n*−6	0.33	0.33	0.30	0.016	0.090
C21:0	0.13	0.12	0.11	0.007	0.384
C20:3 *n*−6	0.24	0.23	0.23	0.011	0.869
C20:4 *n*−6	3.89	3.92	3.73	0.207	0.762
C20:5 *n*−3	1.40 ^a^	1.24 ^ab^	1.19 ^b^	0.139	0.006
C22:5 *n*−3	2.18	2.12	1.89	0.145	0.119
C22:6 *n*−3	0.82 ^a^	0.72 ^b^	0.77 ^ab^	0.132	0.049
∑SFA ^b^	38.54	38.51	39.47	0.936	0.517
∑MUFA ^c^	35.04	35.11	35.73	1.483	0.635
∑PUFA ^d^	25.36	25.37	23.82	0.795	0.123
∑*n*−6 PUFA ^e^	17.79	18.33	17.30	0.520	0.295
∑*n*−3 PUFA ^f^	7.57 ^a^	7.03 ^ab^	6.52 ^b^	0.387	0.002
∑PUFA/∑SFA	0.67	0.66	0.61	0.023	0.154
∑MUFA/∑SFA	0.92	0.91	0.91	0.056	0.940
∑*n*−6 PUFA/∑*n*−3 PUFA	2.38 ^b^	2.63 ^a^	2.67 ^a^	0.078	0.007

^a,b,c^ Means with different superscripts within the same row differ significantly (*p* < 0.05). ^a^ CLA—conjugated linoleic acid. ^b^ SFA = C14:0 + C15:0 + C16:0 + C17:0 + C18:0 + C20:0 + C21:0 + C22:0 + C24:0. ^c^ MUFA = C14:1 *n*–5 + C16:1 *n*−7 + C18:1 *n*−7 + C18:1 *n*−9 + C20:1 *n*−9. ^d^ PUFA = *n*−6 PUFA + *n*−3 PUFA. ^e^
*n*−6 PUFA = C18:2 *n*–6 + C18:3 *n*−6 + C20:2 *n*−6 + C20:3 *n*−6 + C20:4 *n*–6 + C22:4 *n*−6. ^f^
*n*−3 PUFA = 18:3 *n*−3 + C20:3 *n*−3 + C20:4 *n*−3 + C20:5 *n*−3 + C22:5 *n*−3 + C22:6 *n*−3.

**Table 6 foods-09-01255-t006:** Sensory characteristics (scores from 0 to 100) and technological properties of fallow deer *longissimus lumborum* muscle, after the deer were fed different diets.

	Diet				
	Pasture (P) LSM	Barley (B) LSM	Lysine (BL) LSM	SEM	*p*-value
Aroma intensity	62.2	61.7	62.2	2.76	0.957
Game aroma intensity	60.5	62.4	59.1	2.49	0.268
Tenderness	64.9	62.5	62.6	3.47	0.472
Juiciness	57.2	58.5	61.5	3.36	0.114
Flavour intensity	62.8	62.0	63.4	3.23	0.712
Game flavour intensity	60.8	60.8	60.6	2.89	0.988
Grassy flavour	50.6 ^a^	45.4 ^b^	43.9 ^b^	2.43	0.008
Astringency flavour	41.6	42.3	43.1	3.01	0.769
Liver flavour	45.0 ^b^	52.3 ^a^	52.1 ^a^	2.90	0.003
Thawing loss (%)	4.9 ^b^	7.7 ^a^	7.4 ^a^	0.38	<0.001
Cooking loss (%)	18.7	18.5	20.0	0.86	0.406

^a,b,c^ Means with different superscripts within the same row differ significantly (*p* < 0.05).

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
