# Peer review of "The Effect of Barley and Lysine Supplementation on the longissimus lumborum Meat Quality of Pasture-Raised Fallow Deer (Dama dama)"

_foods, 2020, doi:10.3390/foods9091255_

Round 1

Reviewer 1 Report

Major comments:

* The number of references (56) makes no sense in such a short manuscript. Half of the cites are redundant and useless and should be deleted.

RESULTS:

* L.135-139. Did the authors consider the water losses related to all these procedures? Could those water losses be implicated in the results of the sensory analysis?

* L. 172-178 and Table 3. Please, add for each effect, the (%) of difference (statistically significant and tendencies).

-The results for “soluble collagen” were not explained.

-The P-value for “total collagen” in table 3 is different than the one described in the text.

- Please, review the values and statistical analysis of “Moisture”. Considering the SEM value (Table 3), I agree with the fact that P and BL were different but not sure about B (considering N = 15).

* L. 184-189 and Table 4. Please, review the P-values and letter assignment of Table 4. Methionine and Aspartic acid showed P < 0.05, Phenylalanine, Cysteine and Serine P = 0.05 and Isoleucine P = 0.072. All of them must be described in the results section and letters assessed.
Besides, L 184-189 must be rewritten accordingly.
Asparagine and Glutamine were not described in Table 4.
Please, add for each effect, the (%) of difference (statistically significant and tendencies).

* The authors have not provided the FA profile of the diets so it is impossible to assess the content of Table 5 and L.190-198.

Please, add for each effect, the (%) of difference (statistically significant and tendencies).

* L. 200-205. Please, add for each effect, the (%) of difference (statistically significant and tendencies).

Besides, P-values of L. 204 and 205 are different than those shown in Table 6.
L. 205-207. This comment belongs to the discussion, not to the results.

DISCUSSION:

-L. 224-254. These paragraphs must be rewritten. The discussion must be focussed on actual statistical differences and numerical tendencies.
For example, there is no numerical tendency in IMF between B and BL with that letter assignment. If the authors want to discuss those differences, they should consider the use of another statistical procedure (Duncan or Bonferroni test?).

-L. 267-270. Slight increase? Meaning? Please, use a more accurate explanation. In fact, the authors stated that further investigation would be required to understand the effects so this is a good opportunity to describe, analyze and discuss the results more in-depth!

-L. 271-286. Include in the discussion references such as [9] (among others)… A lot of differences in the FA profile between pasture and concentrate have been described in the literature. Please, compare and discuss.

-L. 293-320. It seems that even the authors expected more statistical differences in the sensory analysis. With the differences in FA profile, AA profile and collagen and IMF contents, I would also expect more statistically significant variables in Table 6.
The authors should consider studying the effect of moisture (statistically significant, table 3) and the water losses during vacuum storage and thawing.

Minor comments:

- Please, include the value of the N in each table.

-L.122-133. Please, describe more precisely the anatomical location (cranial, medial or caudal); the LL muscle is heterogeneous.

-L. 170. Delete the word “results”.

-L. 186. contents

-L. 192. were reported

Reviewer 2 Report

Review of paper by Bureš et al.: The effect of barley and lysine supplementation on 2 the longissimus lumborum meat quality of pasture-3 raised fallow deer (Dama dama) (Manuscript number: Foods - 909662)

The aim of the experiment was to compare the meat quality (chemical composition, amino acid composition, fatty acid profile and sensory characteristics) of fallow deer in grass-feeding system and supplemented it with barley and lysine. The subject of manuscript falls within the general scope of the journal. It is well written and contains new scientific results.

General remarks:

I know it is difficult to study the feeding of grazing animal species. Nevertheless, it would be useful to study the composition of plant fauna. This would be important for estimating nutritional value and to determine aromatic plants that affect the taste of meat.

In several cases, it would have been useful to compare the results with other grazing ruminant species.

Authors wrote “However, the underlying cause of the effect of feeding RPAAs on AA composition in meat is not clear, and requires further investigation.” I think it would be useful to compare your results with more other animal species how lysine supplementation affects amino acid composition. In the absence of the data referred to in the first paragraph, it may be that no lysine supplementation was required.

Detailed remarks:

L37: Based on the title of the article, it is only about wild boar, although it may contain more information about game meat.

L44-45: Please, be more specific. Other livestock species include poultry and pigs (fed with cereals) and sheep and cattle for fattening (grazing ruminants).

L108-110: Whole or part of the LL, which part?

L170: Please, delete “3. Results”.

L178 and Table 3: Please, check which of them is correct (0.087 or 0.077).

Table 3: In case of moisture, crude protein and soluble collagen, it is enough to write one decimal.

L190-198: I suggest writing a sentence about the health benefit of EPA and DHA.

L239-248: I think, the allometric growth of different tissues could be the most important factors influencing the results.

Table 6: One decimal for means and two decimals for SEM are enough.

Conclusions are the repetition of the main results. Please, rewrite it.

Round 2

Reviewer 1 Report

The authors have taken into account all the comments. The quality of the manuscript has increased exponentially.

Minor comments:

  • 175. Please, delete the number “3.”
  • 261. It should say “contents” to agree with L. 262. “were”.
    L. 268. …findings [6].
    L. 268-269. …no influence neither on total collagen content nor collagen solubility in FD meat.
    And so on…
    Please, review the whole text to correct similar mistakes.
  • Please, check the reference list for some journals wrongly abbreviated:
    407, 454. Please, check if the abbreviation for “Small Ruminant Research” is “Small Ruminant Res.”
    L. 414. Anim. Feed Sci. Tech. ?
    L. 400, 450, 479, 544. J. Sci. Food Agr. ?
    L. 539, 548. J. Agr. Food Chem. ?
